Ploidy level enhances the photosynthetic capacity of a tetraploid variety of Acer buergerianum Miq.

Wang Yi 1 2
Jia Bingyu 1 3
Ren Hongjian 4
Feng Zhen 1 fengzn408@qq.com
1 College of Forestry, Key Laboratory of State Forestry Administration for Silviculture of the Lower Yellow River, Shandong Agricultural University , Tai’an, Shandong Province , China
2 Laboratory of Systematic Evolution and Biogeography of Woody Plants, School of Ecology and Nature Conservation, Beijing Forestry University , Beijing , China
3 Forestry Bureau of Huguan County , Changzhi, Shanxi Province , China
4 Forestry Protection and Development Center of Ningyang County, Ningyang , Tai’an, Shandong Province , China
Lambrughi Matteo
Electronic publication date: 2021 Dec 16
Publication date: 2021
Volume: 9
Electronic Location ID: e12620
Received 2020 Aug 25; Accepted 2021 Nov 18
Copyright: © 2021 Wang et al.
Copyright year: 2021
Copyright holder: Wang et al.
License: This is an open access article distributed under the terms of the Creative Commons Attribution License, which permits unrestricted use, distribution, reproduction and adaptation in any medium and for any purpose provided that it is properly attributed. For attribution, the original author(s), title, publication source (PeerJ) and either DOI or URL of the article must be cited.
License URL: https://creativecommons.org/licenses/by/4.0/

Keywords: Acer buergerianum, Autotetraploid, Transcriptome, Photosynthetic capacity, Chloroplast ultrastructure

Funding: This work was supported by the Shandong Agricultural Seeds Engineering Project Improvement and Demonstration of New Varieties with High Quality and High Yield for Medicinal Trees— Acer truncatum (2016LZGC014) and Tai’an science and technology innovation development project (the category of Policy guidance) “ Research on application technology of Acer afforestation in barren mountain” (2020NS065). The funders had no role in study design, data collection and analysis, decision to publish, or preparation of the manuscript This work was supported by the Shandong Agricultural Seeds Engineering Project: Improvement and Demonstration of New Varieties with High Quality and High Yield for Medicinal Trees—Acer truncatum (2016LZGC014) and Tai’an science and technology innovation development project (the category of Policy guidance) “Research on application technology of Acer afforestation in barren mountain” (2020NS065). The funders had no role in study design, data collection and analysis, decision to publish, or preparation of the manuscript.

==============================
Background

Polyploidy plays an important role in plant breeding and has widespread effects on photosynthetic capacity. To determine the photosynthetic capacity of the tetraploid variety Acer buergerianum Miq. ‘Xingwang’, we compared the gas exchange parameters, chloroplast structure, chlorophyll contents, and chlorophyll fluorescence parameters between the tetraploid Acer buergerianum ‘Xingwang’ and the diploid ‘S4’. To evaluate the effects of genome duplication on the photosynthetic capacity of Acer buergerianum ‘Xingwang’, the transcriptomes of the autotetraploid ‘Xingwang’ and the diploid ‘S4’ of A. buergerianum were compared.

Methods

The ploidy of Acer buergerianum ‘Xingwang’ was identified by flow cytometry and the chromosome counting method. An LI-6800 portable photosynthesis system analyzer was used to assess the gas exchange parameters of the tetraploid variety ‘Xingwang’ and diploid variety ‘S4’ of A. buergerianum. We used a BioMate 3S ultraviolet-visible spectrophotometer and portable modulated fluorometer to measure the chlorophyll contents and chlorophyll fluorescence parameters, respectively, of ‘Xingwang’ and ‘S4’. Illumina high-throughput sequencing technology was used to identify the differences in the genes involved in the photosynthetic differences and determine their expression characteristics.

Results

The single-cell DNA content and chromosome number of the tetraploid ‘Xingwang’ were twice those found in the normal diploid ‘S4’. In terms of gas exchange parameters, the change in stomatal conductance, change in intercellular CO2 concentration, transpiration rate and net photosynthetic rate of ‘Xingwang’ were higher than those of the diploid ‘S4’. The chlorophyll contents, the maximal photochemical efficiency of PSII and the potential photochemical efficiency of PSII in ‘Xingwang’ were higher than those of ‘S4’. The chloroplasts of ‘Xingwang’ contained thicker thylakoid lamellae. By the use of Illumina sequencing technology, a total of 51,807 unigenes were obtained; they had an average length of 1,487 nt, and the average N50 was 2,034 nt. The lengths of most of the unigenes obtained ranged from 200–300 bp, with an average value of 5,262, followed by those longer than 3,000 bp, with an average value of 4,791. The data revealed numerous differences in gene expression between the two transcriptomes. In total, 24,221 differentially expressed genes were screened, and the percentage of differentially expressed genes was as high as 46.75% (24,224/51,807), of which 10,474 genes were upregulated and 13,747 genes were downregulated. We analyzed the key genes in the photosynthesis pathway and the porphyrin and chlorophyll metabolism pathway; the upregulation of HemB may promote an increase in the chlorophyll contents of ‘Xingwang’, and the upregulation of related genes in PSII and PSI may enhance the light harvesting of ‘Xingwang’, increasing its light energy conversion efficiency.

Introduction

Polyploidy plays an important role in plant breeding (Otto, 2007; Dhooghe et al., 2011) because polyploid plants often more strongly express the characteristics targeted by breeders—including traits related to morphology, metabolism, genetic adaptation ability and tolerance—than their diploid counterparts (Allario et al., 2011; Comai, 2005; Leal-Bertioli et al., 2017; Van Laere et al., 2010). Polyploid plant growth may require more energy and carbohydrates (Evans, 2013), which are primarily obtained from the process of photosynthetic carbon assimilation in the leaves; therefore, the enhanced photosynthetic capacity of polyploids compared to diploids may be advantageous (Evans, 2013). Photosynthetic capacity is related to the chlorophyll content, the structure of chloroplasts, and related gene expression resulting from the polyploidization event. In terms of chlorophyll contents and the structure of chloroplasts, chloroplasts contain chlorophyll, which is the site of photosynthesis, so the effect of chlorophyll contents and the structure of chloroplasts on photosynthetic capacity is very obvious (Eckhardt, Grimm & Hörtensteiner, 2004). Previous reports showed that tetraploids of C. nankingense had significantly higher chlorophyll a and b contents than diploids and increased photosynthetic capacity (Dong et al., 2017). In an ultrastructural study of polyploid chloroplasts and their characteristics at the transcriptome level, Chen et al. (2010) found that the number of chloroplasts, grana and lamellae in heterotetraploids of Cucumis was higher than that in diploids. Zhu et al. (2012) also found that the chloroplasts, grana and lamellae in tetraploids were present in greater numbers than in diploids during their comparison of the leaf ultrastructure of watermelon seedlings with different ploidy. In previous studies on the ultrastructure of chloroplasts, changes in the plastoglobuli in chloroplasts were ignored. The so-called plastoglobuli are also called plastid globules and lipidosomes because of their storage of fat (Wang, 2000). The plastoglobule in the chromoplast may be the product of the disintegration of the thylakoid membrane. With the disintegration of thylakoids, the number of plastoglobuli increases, and therefore, this number may be related to whether the thylakoid membrane structure is stable (Greenwood, Leech & Williams, 1963). The thylakoid membrane contains photosynthetic pigments and a variety of protein molecules and undergoes reactions such as those involving light energy absorption, transfer and transformation, and electron transfer; therefore, the number and structure of thylakoids are important factors that influence photosynthesis (Allen & Forsberg, 2001; Liu et al., 2012). Dong et al. (2017) found that whole genome duplication enhances the photosynthetic capacity of C. nankingense, mainly because the number of genes associated with chlorophyll synthesis was upregulated in the tetraploid form, while those associated with chlorophyll degradation were downregulated.

The photosynthetic rate of plant cells is related to the DNA content of the cell itself; thus, the increased DNA content in polyploid mesophyll cells may endow them with a higher photosynthetic rate (Coate et al., 2012; Warner & Edwards, 1993). However, for different plants, the number of mesophyll cells per unit area changes with the increase in cell volume (Warner & Edwards, 1993); thus, at the leaf level, the effect of polyploidy on the photosynthetic capacity will vary from species to species. Therefore, in some plants, the photosynthetic rate of tetraploids is higher than that of diploids, such as Paulownia australis (Wang et al., 2013). Studies have also shown that the photosynthetic rate of some tetraploids is lower than that of their diploid counterparts, as is the case for Fragaria (Gao et al., 2017).

Acer buergerianum belongs to the Sapindaceae family, and it is native to eastern China (Hayashi, 2007) and Japan (Matsue & Takeda, 2009). In China, Acer buergerianum is mainly distributed across the middle and lower reaches of the Yangtze River and the Taiwan region (Chinese Flora Editorial Board of the Chinese Academy of Sciences, 1981). Acer buergerianum is highly adaptable, drought-resistant, and has excellent wood. It is also an important color-leaf tree species and forest landscape resource (Feng et al., 2012; Guo et al., 2013). Therefore, vigorously developing Acer buergerianum is of great interest for improving the ecological environment and improving people’s living standards. However, research on Acer buergerianum is mainly focused on landscape applications, and there is less research on its polyploid photosynthetic capacity and chloroplast ultrastructure and on the transcriptomic analysis of photosynthesis in this species, which limits its development. The chromosome cardinality of A. buergerianum is 13, the number of diploid chromosomes is 2n = 2x = 26 (Xu, 1998), and the number of tetraploid chromosomes is 2n = 4x = 52 (Xu, 1998). Acer buergerianum ‘Xingwang’ is a tetraploid and is propagated by grafting. In our previous research, the tetraploid ‘Xingwang’ showed a significant advantage over diploid plants in terms of vegetative growth, including leaf length, leaf width, and leaf thickness (An et al., 2018). Since the increased carbohydrate accumulation required for enhanced vegetative growth is closely associated with more effective photosynthesis (Warner & Edwards, 1993), vegetative growth superiority might indicate improved photosynthetic performance in the tetraploid Acer buergerianum.

To reveal the photosynthetic capacity of the tetraploid Acer buergerianum ‘Xingwang’, we compared the differences between the gas exchange parameters, chlorophyll contents, chlorophyll fluorescence parameters and chloroplast ultrastructure of the diploid ‘S4’ and the autotetraploid ‘Xingwang’. Additionally, comparisons between the transcriptomes of the autotetraploid ‘Xingwang’ and its diploid relative regarding photosynthesis were performed using Illumina high-throughput sequencing technology. This study lays a foundation for an in-depth understanding of the tetraploid variety A. buergerianum ‘Xingwang’ and the genetic engineering of this species.

Materials and Methods

Plant materials

The autotetraploid ‘Xingwang’ (X) originated from the colchicine-induced somatic chromosome doubling of the diploid ‘S4’ (S), and 20 clones with genetic consistency were obtained by grafting propagation. Young leaves of six-year-old ‘Xingwang’ and autodiploid ‘S4’ were used as experimental materials. All the test materials were obtained from the gardens of Shandong Agricultural University Forestry College (N36°10′ 12.44″, E117°09′ 40.33″) and were grown outdoors. The region has a temperate monsoon climate with an average annual temperature of 13 °C, average annual precipitation of 688.3 mm, and annual average relative humidity of 66%. The soil is loamy.

Polyploidy identification

The single-cell DNA content of the polyploid X was determined by flow cytometry (Handyem HPC-150, Canada) (Zhang, Guo & Deng, 2006). Normal leaves of X and S were collected; the leaves were washed, and the veins were removed. Then, 20 mg of the remaining leaf tissue was placed in a culture dish, and fresh lysis solution was added. A double-sided blade was used to chop the leaf tissue quickly in a Petri dish, and the tissue was processed for 3–5 min. A pipette was used to extract 1 ml of the chopped sample, which was filtered into a 1.5 ml centrifuge tube with a 300-mesh nylon sieve. Then, 1 mg/ml propidium iodide solution was added to the centrifuge tube to reach a final propidium iodide concentration of 0.1 mg/ml, and the solution was allowed to stand for 5 min. The processed solution was filtered through a 300-mesh nylon sieve again into a new centrifuge tube and then tested on the machine. Three replicates were employed for each test sample from the same plant, and at least 9,000 cells were collected each time. The lysate formulation was 0.1 mol/l citric acid, 0.5% TWEEN-20, 0.4 mol/l disodium hydrogen phosphate, and 1% PVP-10 (Kron & Husband, 2012).

Chromosome ploidy identification of the polyploid X was performed using the chromosome counting method (Zhang et al., 2005). The innermost 0.5 cm length of the stem tip was selected as the experimental material, which was treated at −20 °C for 18 h in a stable solution (anhydrous ethanol:glacial acetic acid:trichloromethane = 5:3:2) and then stored in 75% alcohol at 4 °C. The material was removed and washed 3 times in ultrapure water. The excess water was absorbed with clean filter paper and the sample was transferred to 0.075 mol/l KCl solution to reach hypotonicity for 30 min. Following dissociation, the material was rinsed with ultrapure water 3 times and placed in ultrapure water for 30 min to become hypotonic. The water and Carbo Fuchsin dye (Solarbio, Tongzhou, China) solution were then absorbed for 15–20 min, the material was pressed into a tablet, and 10–20 relatively clear metaphase cells were found under the microscope for identification.

Gas exchange parameters

The experimental site is in the garden of Shandong Agricultural University Forestry College (the environmental conditions are the same as those detailed above). We chose a sunny day in the middle of each month from May to September 2019 and selected tetraploid X and diploid S plants of equal size. We used an LI-6800 portable photosynthesis system (LI-COR, Lincoln, NE, USA) to measure the net photosynthetic rate (Pn), stomatal conductance (Gs), transpiration rate (Tr), and intercellular carbon dioxide concentration (Ci) of the third pair of sun-exposed, physiologically mature leaves from the top of the side branch to the trunk at the same orientation from three plants, with three leaves collected from each plant. The gas exchange parameter data were recorded every 2 h from 8:00 to 18:00, and the average value was used for the processing analysis.

Determination of chlorophyll contents and chlorophyll fluorescence parameters

Fresh leaves from diploid and tetraploid plants were cut into pieces and submerged in 80% (v/v) acetone in the dark at room temperature to extract their chlorophyll (Arnon, 1949). The chlorophyll a (Chl a), chlorophyll b (Chl b) and total chlorophyll (a+b) contents were measured using a BioMate 3S ultraviolet-visible spectrophotometer (Thermo Fisher Scientific Inc., Waltham, MA, USA) at 470, 649, and 665 nm according to the methods of Li (2000).

The chloroplast fluorescence parameters of the X and S leaves were measured under slightly excessive light stress. The control group was exposed to a natural light intensity (88,000 lx) while the experimental group was exposed to a 20% increased light intensity (105,600 lx). The chlorophyll fluorescence parameters were measured using a portable modulated fluorometer (PAM-2500; Heinz Walz, Pfullingen, Germany). The maximal fluorescence (Fm) and the minimal fluorescence (Fo) were recorded. The maximal photochemical efficiency of PSII (Fv/Fm) and the potential photochemical efficiency of PSII (Fv/Fo) were calculated as Fv/Fm = (Fm − Fo)/Fm and Fv/Fo = (Fm − Fo)/Fo (Kramer et al., 2004).

Nine samples from three individuals of each ploidy type were measured.

Electron microscope sample preparation of leaf chloroplasts

The mature leaves on the annual sun-exposed branches were collected at 4 °C, and the container and instruments were precooled. Fresh leaf samples (1 × 1 mm) were cut from the middle part of the leaves along the main vein, fixed in 2.5% glutaraldehyde fixative solution, and stored at 4 °C. The fixed leaves were washed 3 times with 0.1 mol/l phosphate buffer solution containing 1% osmium acid (Pelco, EM level, Fresno, CA, USA) at 4 °C for 3 h, washed 3 times with buffer solution, and then dehydrated step by step with ethanol and propylene oxide. A replacement was performed; then, the embedding material was impregnated with Spurr resin (Spi-Chem, West Chester, PA, USA). Last, the samples were polymerized in an oven at 70 °C. The embedding blocks of the different materials were placed on an ultrathin microtome (EM UC6; Leica, Wetzlar, Germany) for sectioning. The thickness of the ultrathin sections was 70 nm. The sections were stained with uranyl acetate (Spi-Chem, ACS level, West Chester, PA, USA) and lead citrate trihydrate (Spi-Chem, ACS level, West Chester, PA, USA) and observed and photographed under a transmission electron microscope (JEM1230; JEOL, Akishima, Tokyo, Japan). Image-Pro Plus 6.0 (Media Cybernetics, Inc., Rockville, MD, USA) was used to produce statistics on the thylakoid lamella thickness. Three samples of X and S were analyzed, and for each sample, the thylakoid lamella thickness and number of plastoglobules from at least three chloroplasts were detected.

RNA extraction, library construction and RNA-seq

During the transcriptome sequencing process, for X and S, three biological replicates were taken from the same plant, and they were denoted X1, X2, and X3 and S1, S2, and S3. Tender leaves from the annual sunny branches were obtained, and an Adelaide Biologicals EASYspin Plant RNA Quick Extraction Kit was used to extract the total RNA from each X and S sample. A spectrophotometer (Thermo Fisher Scientific, Inc., Waltham, MA, USA) was used for RNA quality inspection, and the integrity of the RNA was confirmed by 1% agarose gel electrophoresis.

Oligo (dT) magnetic beads were used to enrich the mRNA with a polyA tail, and interrupting buffer was used to fragment the resulting RNA. Random N6 primers were used for reverse transcription, and two strands of cDNA were synthesized to form double-stranded DNA. The ends of synthetic double-stranded DNA were filled in, and the 5′ end was phosphorylated. The 3′ end forms a sticky end that protrudes with a base ‘A’; then, there is a complementary end with a protruding base ‘T’ on the 5′ end, and the ligated product was amplified by PCR with specific primers. The PCR products were thermally denatured into single-stranded DNA, and then the single-stranded DNA was circularized with a bridge primer to obtain a single-stranded circular DNA library. Finally, the BGISEQ-500 platform was used to obtain raw sequencing data.

D. novo assembly and functional annotation

The software SOAPnuke (Cock et al., 2010) was used to filter the raw data for sequencing, and the filtered data were called clean reads and saved in FASTQ format. Trinity software (Grabherr et al., 2011) was used to assemble then process clean reads in turn. Trinity has three independent modules: Inchworm, Chrysalis and Butterfly. The data were first assembled with Inchworm to obtain the full-length isoform transcript data and the fragment form of the isoform-containing transcript. Next, in the Chrysalis module, the contigs that may have variable splicing and parallel genes are clustered to construct a large number of individual de Bruijn maps. Last, the linear path is merged with continuous nodes in the de Bruijn maps in the Butterfly module and full-length transcript splicing subtypes are extracted.

Then, TGICL software (Pertea et al., 2003) was used to cluster the transcripts to remove redundancies and obtain the unigenes. After clustering, the unigenes were divided into two classes: clusters with the prefix CL and singletons with the prefix Unigene.

For the resulting transcript data, we used Trinity to perform open reading frame (ORF) predictions, and we annotated the transcript with complete ORFs. After assembling the unigenes, we performed bioinformatics analysis and aligned the predicted protein sequences to six functional databases, namely, NCBI nonredundant protein (Nr), NCBI nucleotide in sequences (NT), Swiss-Prot, Pfam, Kyoto Encyclopedia of Genes and Genomes (KEGG), euKaryotic Orthologous Groups (KOG), and Gene Ontology (GO), for annotation and classification. The best alignment results were used to annotate the transcriptome sequences and to determine the orientation of the sequences. The All-unigene results aligmented to the NR database were annotated to the GO database via Blast2GO software (Conesa et al., 2005), and the metabolic pathways were compared with the KEGG database (Kanehisa et al., 2008).

Differential gene analysis

The DEGseq method (Wang et al., 2010) was used to detect differentially expressed genes (DEGs), calculate the p-values (p-value, hypothesis test probability) and correct the p-values to Q-values. To improve the accuracy of the DEGs, we implemented the fold change as more than twice (fold change ≥ 2.00) and the Q-values (adjusted p-value) ≤ 0.05 to screen differential genes and detect the DEGs based on the principle of a negative binomial distribution. The DEGs related to photosynthetic characteristics were selected for analysis.

Real-time quantitative RT-PCR (qRT-PCR) verification

To confirm the reliability of the sequencing analysis, the expression of the candidate genes was measured by qRT-PCR. First, ABM’s 5X All-In-One RT MasterMix Kit (with the AccuRT Genomic DNA Removal Kit) was used to reverse transcribe the RNA into the first strand of cDNA, and the product was stored at −20 °C. Nine candidate genes were selected, the sequences of the target genes were obtained from the transcriptome results (Table S1), and primers were designed. TUB was used as the internal reference gene, and a Bio-Rad CFX96TM Real-Time PCR instrument was used to complete the qRT-PCR. The reaction procedure was as follows: 95 °C for 30 s; 39 cycles of 95 °C for 5 s and 60 °C for 30 s; and then 95 °C for 10 s. The qRT-PCR results were calculated using the 2−ΔΔt method, and whether the expression trend of the transcriptome results was consistent with the verification results was assessed. The sequences of the primers used are shown in Table S2.

Statistical analysis

Descriptive statistical analyses were used for the measured parameters to obtain the means and standard errors (SEs). One-way analysis of variance (ANOVA) was used to analyze the significance of the differences in the gas exchange parameters, chlorophyll contents, chlorophyll fluorescence parameters and chloroplast ultrastructure between X and S, with the ploidy level as a fixed factor. The significance level was set at p < 0.05. Statistical analyses were performed with SPSS 16.0 (SPSS, Inc., Chicago, IL, USA).

Results

Identification of the A. buergerianum tetraploid

By observing the stem tip under the microscope, we found that the chromosome number of X was significantly higher than that of S by twofold (Fig. S1).

The DNA content in a single cell was detected by flow cytometry, and the results are shown in Fig. S2. The ploidy level of the plant can be inferred from the peak figure of its relative DNA content (Doležel, Greilhuber & Suda, 2007; Paul & Husband, 2012). The main peak single-cell DNA content in the S leaves was at channel 100, while the main peak single-cell DNA content in X leaves was at channel 200. Thus, the single-cell DNA content of X was twice that of S; therefore, X was determined to be tetraploid.

Gas exchange parameters, chlorophyll contents and chlorophyll fluorescence parameters

Similar patterns of diurnal changes in Tr were observed between diploid and tetraploid A. buergerianum X and S leaves from May to September (Fig. S3). In May, July, and September, a single-peak curve was observed that first increased and then decreased, reaching a maximum at approximately 12:00. The peak Tr of X was higher than that of S, the Tr reached the maximum in July, the maximum Tr of X was 1.33 mol·m−2·s−1, and the maximum Tr of S was 1.23 mol·m−2·s−1. In June and August, the daily variation of the whole day showed a double-peak curve, with the first peak appearing at approximately 10:00 and the second peak appearing at 14:00. In X and S, the first peak was greater than the second peak and fell into a valley at approximately 12:00.

The daily variation trends in the Pn of X and S leaves from May to September were similar (Fig. S4). In May, July, and September, a single-peak curve first increased and then decreased. Starting at 8:00, the Pn gradually increased, reached a maximum at approximately 12:00, and then gradually decreased; the Pn reached its maximum in August. The peak Pn of X was higher than that of S. The maximum Pn of X was 11.15 µmol·m−2·s−1, and the average Pn was 6.63 µmol·m−2·s−1. The maximum Pn of S was 8.56 µmol·m−2·s−1, and the average Pn was 5.72 µmol·m−2·s−1. The diurnal changes in June and August showed a double-peaked curve. The first peak appeared at approximately 10:00, and the second peak appeared at 14:00. The first peaks of X and S were both higher than the second peak, and the values fell into a valley at approximately 12:00. Over the 5 months from May to September, the maximum Pn of X increased by 4.31%, 12.35%, 8.48%, 30.31%, and 7.76% over S.

The diurnal variation in Gs of X and S leaves from May to September was similar, and the Gs of X during different months was higher than that of S (Fig. S5). X and S showed a single-peak curve that first increased and then decreased in May, July, and September and was lower at 8:00 and 18:00. Starting from 8:00, Gs gradually increased and reached a maximum at approximately 12:00. The diurnal changes in June and August showed a bimodal curve. The peak Gs of X was higher than that of S, and the Gs of X and S reached their maximum levels in August. In June, the first peaks of X and S both appeared at approximately 10:00, and the second peak appeared at 14:00. The first peaks of X and S both appeared at approximately 10:00 in August, and the second peaks appeared at approximately 14:00 for X and at approximately 16:00 for S. The first peaks of X and S were both larger than the second peaks, and their values fell into a valley at approximately 12:00.

From May to September, the daily change curve in Ci between X and S cells presented a single valley change (Fig. S6). The diurnal variation curves of X and S both first decreased and then increased, and the Ci of X was greater than that of S during different months. Starting from 8:00, as the environmental conditions changed, the Ci gradually decreased, and the lowest value appeared at approximately 12:00 and then gradually increased.

Among the gas exchange parameters of tetraploid X and diploid S, Tr, Pn, Gs and Ci all exhibited extremely significant differences (P < 0.01). From May to September, the maximum values of all the gas exchange parameters of X were higher than those of S (Table 1).

Table 1 Maximum gas exchange parameters of different ploidies.

It contains the maximum values of Tr, Pn, Ci, Gs from May to September.

Month	Ploidy	Maximum net photosynthetic rate (Pn max, µmol·m−2·s−1)	Maximum stomatal conductance (Gs max, mol·m−2·s−1)	Maximum CO2 concentration (Ci max, µmol·mol−1)	Maximum
transpiration rate
(Tr max, mol·m−2·s−1)	
May	Tetraploid X
Diploid S	5.3227 ± 0.116a
5.1029 ± 0.020b	0.1192 ± 0.010a
0.0864 ± 0.004b	309.7800 ± 5.34a
293.4500 ± 3.83b	1.1123 ± 0.128
0.9872 ± 0.036	
June	Tetraploid X
Diploid S	5.9249 ± 0.153a
5.2735 ± 0.030b	0.1174 ± 0.008a
0.0828 ± 0.000b	351.7200 ± 3.112a
301.0400 ± 16.64b	1.2764 ± 0.230
1.0885 ± 0.046	
July	Tetraploid X
Diploid S	8.4846 ± 0.179a
7.8211 ± 0.029b	0.1123 ± 0.006a
0.0907 ± 0.000b	355.2300 ± 8.338a
301.0400 ± 16.64b	1.3319 ± 0.110
1.2282 ± 0.006	
August	Tetraploid X
Diploid S	11.1538 ± 0.455a
8.5592 ± 0.024b	0.1900 ± 0.143a
0.1454 ± 0.005b	335.2800 ± 2.656a
301.4300 ± 10.334b	1.1238 ± 0.048
0.8357 ± 0.011	
September	Tetraploid X
Diploid S	9.1703 ± 0.180a
8.5103 ± 0.032b	0.1448 ± 0.006a
0.1189 ± 0.004b	323.5100 ± 6.417a
294.5100 ± 11.461b	1.2562 ± 0.030
0.9756 ± 0.014	
Notes:

Values represent the mean ± standard error (SE) of at least three tagged in dividuals of each ploidy type.

Different lowercase letters within a row indicate statistically significant differences (p < 0.01) between the two ploidy level on the same day. The same letters mean the difference is not significant, and the different letters mean the difference is significant.

The analysis of chlorophyll contents showed that the Chl a, Chl b and total chlorophyll contents were significantly higher in X than in S (Fig. 1A). The total chlorophyll contents of X and S were 2.47 mg·g−1 and 1.52 mg·g−1, respectively. The leaves of X contained 38.66% more chlorophyll than those of S. Under slightly excessive light stress, Fv/Fm was 0.77 and 0.83 and Fv/Fo was 3.45 and 4.95 in S and X, respectively, and these values were significantly different (Fig. 1B).

Figure 1 Chlorophyll contents and chlorophyll fluorescence parameters in X and S.

The leaf chlorophyll a/b content and chlorophyll fluorescence parameters in X and S. An asterisk (*) means values for the X and S differ signifcantly according to ANOVA (p < 0.05).

Ultrastructure of the chloroplast

The ultrastructures of X and S mesophyll cell chloroplasts were further compared using a transmission electron microscope (Fig. 2). The chloroplasts of X and S were well developed, with obvious thylakoids and stacked basal particles. However, differences were observed in the internal structure of the chloroplasts from the two varieties. Compared with the chloroplasts in S, the chloroplasts in X contained thylakoid lamellae that were twice as thick (Figs. 2, 3A). In addition, the thylakoid lamellae of X were arranged more closely than those of S, and more starch grains were observed in S. In the chloroplasts of the tetraploid X, the number of plastoglobules was lower than that of the diploid S (Fig. 3B).

Figure 2 Ultrastructure of the mesophyll cell chloroplasts.

(A) and a: A. buergerianum diploid; (B) and b: A. buergerianum tetraploid. Ch, chloroplast; SG, starch grains; G, grana; O, plastoglobuli.

Figure 3 Comparison of thylakoid lamella thickness and plastoglobuli number of X and S.

An asterisk (*) means values for the X and S differ signifcantly according to ANOVA (p < 0.05).

Transcriptome sequencing and assembly

The transcriptomes of X and S produced 42,804,595 and 44,971,637 nt raw data, respectively, and the Q20 percentages were both over 97%.

The unigenes in each sample were clustered to eliminate redundancy, and 51,797 unigenes were obtained, with an average length of 1,487 nt; the average N50 was 2,034 nt.

Transcript database annotation

A total of 44,665 unigenes were annotated and accounted for 86.23% of all the unigenes. The numbers of unigenes annotated in the NR, NT, SwissProt, KEGG, KOG, Pfam, and GO functional databases were 43,076, 36,463, 32,865, 34,715, 34,085, 34,599, and 26,781, respectively (Table S3, Fig. S7).

Analysis of DEGs

A total of 24,221 DEGs (Table S4) were screened between the Acer buergerianum diploid and tetraploid transcriptomes, and they included 10,474 upregulated genes and 13,747 downregulated genes. To study the function of DEGs related to photosynthetic characteristics in diploid and tetraploid plants, GO (Table S5) and KEGG (Table S6) classifications were determined.

We analyzed the pathways of porphyrin and chlorophyll metabolism and photosynthesis. During chlorophyll synthesis in A. buergerianum, five genes encoding HemB were identified, all of which were upregulated. One gene encoded HemC, and its expression was downregulated. One gene encoded HemE, and its expression was downregulated. Three genes encoded CHLH/D/I; one was upregulated, while two were downregulated (Fig. S8, Table 2) (Matsumoto et al., 2004; Beale, 2005).

Table 2 Differentially expressed genes in the chlorophyll synthesis pathway.

It contains information about genename, geneID, and size.

Gene name	Gene ID	Size (bp)	log2 (X/S)	
HemB	CL3197.Contig1	1,743	1.2136	
CL3197.Contig2	1,848	4.9716	
CL3197.Contig3	1,474	1.8448	
CL3197.Contig4	1,213	7.4426	
CL3197.Contig5	1,343	7.2818	
HemC	CL3343.Contig1	1,446	−1.8068	
HemE	CL633.Contig2	1,641	−1.0693	
CHLH/D/I	CL6001.Contig1	2,592	−1.6240	
CL6001.Contig2	2,776	3.7295	
Unigene11625	2,263	−1.1987	

In this study, 58 DEGs were annotated as being related to the photosynthesis pathway, with 30 upregulated genes and 28 downregulated genes. Twelve were photosystem I (PSI)-related genes, of which the PsaA-, PsaB-, and PsaK-related genes were upregulated and the PsaG- and PsaH-related genes were downregulated; 28 were photosystem II (PSII)-related genes, of which the PsbE- and Psb27-related genes were upregulated and the PsbH-, PsbQ-, and PsbS-related genes were downregulated; four were cytochrome b6/f complex-related genes, of which the PetB- and PetC-related genes were downregulated; 10 were photosynthetic electron transport-related genes, of which the PetH-related genes were downregulated; eight were F-type ATPase-related genes, among which the beta-, delta-, and b-related genes were upregulated (Fig. 4) (Yamori & Shikanai, 2016).

Figure 4 Photosynthesis pathway.

Up-regulated genes in A. buergerianum X are represented by purple boxes, and down-regulated genes are represented by blue boxes.

qRT-PCR validation of the candidate DEGs

To verify the reliability of the transcriptome sequencing results, qRT-PCR validation was performed on nine randomly selected candidate genes (Table S1), including four upregulated genes (CL6641.Contig2_All, CL2285.Contig2_All, CL3396.Contig3_All, and CL335.Contig5_All) and five downregulated genes (CL7423.Contig2_All, Unigene5873_All, CL1401.Contig2_All, CL6251.Contig1_All, and CL5835.Contig1_All). The results were consistent with the expression patterns of the transcriptome sequencing results (Fig. S9), validating the transcriptome sequencing results obtained in this study.

Discussion

A comparison of the gas exchange parameters for X and S showed that X plants had a higher Pn, Tr, Gs, and Ci than S plants, which indicates that the gas exchange parameters of the tetraploid form of X were markedly higher those that of the diploid form of S. The Pn of X and S their maximum levels in August, which may be due to favorable conditions for the growth of A. buergerianum such as the photoperiod and light intensity, temperature, and relative humidity (Wang et al., 2006). Among the gas exchange parameters for X and S photosynthesis, the daily variation curves in the Pn and Gs show different changes during different months. The Gs and Ci of tetraploid X were higher than those of S, so the Pn differences between X and S may come from stomatal factors. The increase in the stomatal conductance of X is conducive to the continuous replenishment of CO2 in the intercellular space during photosynthesis, thus maintaining a high intercellular CO2 concentration and lower stomatal limitation (Du et al., 2011); as a result, the Pn of X was significantly higher than that of S. This result was consistent with a previous study showing that stomata from X were significantly larger than those from S (An et al., 2018).

Chloroplasts are the main location for photosynthesis in plants, and their structure directly affects the photosynthesis and characteristics of leaves (Liu et al., 2012). Chen et al. (2010) used diploid melon, allogeneic triploid and allogeneic tetraploid plants as test materials to compare the differences in chloroplast ultrastructure, and the results showed that the number of chloroplasts, basal grains and lamellae increased significantly in the allotetraploids compared to the diploids. However, many previous studies have ignored the changes in thylakoids. In this study, there were more and thicker thylakoid lamellae in X than in S, and the arrangement of thylakoids was more densely stacked in the basal granule, which indicated that their daylighting mechanism was more effective at absorbing and converting light energy (Li et al., 2018). In addition, the number of plastoglobules in X was lower than that in S, which may indicate that the thylakoid membrane structure of X was relatively stable. These differences may be one of the factors that cause the photosynthetic rate of the tetraploid X to be higher than that of the diploid S.

Chlorophyll is the key photosynthetic pigment in higher plants (Masuda & Fujita, 2008). When plants are deficient in chlorophyll, abnormal thylakoid structure development often occurs (Falbel & Staehelin, 2008). The synthesis and degradation of chlorophyll affect the development of chloroplasts and further affect the photosynthesis of plants (Eckhardt, Grimm & Hörtensteiner, 2004). Here, the chlorophyll content of X was significantly increased compared with that of S, which was also an important factor that made the photosynthetic capacity of X significantly stronger than that of S.

The increase in chlorophyll content in X was presumably associated with the upregulation of the chlorophyll synthesis genes HemB caused by whole genome duplication. Green plants need to complete 16 steps to synthesize chlorophyll; the synthesis of chlorophyll b involves four processes, 5-aminolevulinic acid (ALA) synthesis, protoporphyrin synthesis, the conversion of protoporphyrin to protochlorophyll, and the conversion of protochlorophyll to chlorophyll a and b (Fig. S8) (Ohmiya et al., 2014). Here, among the genes involved in chlorophyll synthesis, HemB in X underwent major changes, and all five genes involved in HemB were upregulated. The expression of HemB (which encodes aminolevulinate dehydratase, ALAD) has a positive regulatory effect on chlorophyll biosynthesis (Tang et al., 2012); therefore, the upregulation of the gene encoding HemB may result in increased chlorophyll synthesis, thereby promoting the photosynthesis of tetraploids. This finding is consistent with the significantly higher chlorophyll contents in X than in S.

Research has shown that the genes encoding the components of PS II and PS I are associated with dosage sensitivity (Coate et al., 2014). In many cases, gene dose sensitivity correlates with gene duplication, as described by the gene balance theory (Edger & Pires, 2009). During photosynthesis, psaA and psaB are mainly related to light harvesting (Chitnis, 1996); thus, the upregulation of their related genes may enhance the light harvesting of X. In addition, the differential expression of the related genes in PS II may promote the efficiency of solar energy conversion into chemical energy. This hypothesis was confirmed by measurements of the X and S fluorescence parameters. Under mild light stress, the Fv/Fo and Fv/Fm of X were significantly higher than those of S, indicating that the autotetraploid had higher light harvesting and conversion efficiency than its diploid (Liang, Dou & Feng, 2004). The chloroplast ATPase on the thylakoid membrane is responsible for catalyzing light-driven ATP synthesis and providing energy for carbon fixation in photosynthesis (Zhang et al., 2018); therefore, the upregulation of related genes in ATPase may provide more energy for carbon fixation in photosynthesis, thereby increasing the photosynthetic efficiency of polyploid X and meeting its growth needs. Finally, we present a schematic diagram for the possible mechanisms to show the reasons for the increased photosynthetic capacity of X (Fig. 5).

Figure 5 Schematic diagram for the possible mechanisms for the increased photosynthetic capacity of X.

Conclusions

In this study, ploidy identification of the variety Acer buergerianum ‘Xingwang’ confirmed that it is a tetraploid. The gas exchange parameters, chlorophyll contents, Fv/Fo and Fv/Fm of the tetraploid ‘Xingwang’ were significantly higher than those of the diploid ‘S4’. Moreover, ‘Xingwang’ has a thicker thylakoid lamella and fewer plastoglobules than ‘S4’. These reasons cause ‘Xingwang’ to have a higher photosynthetic capacity than ‘S4’. The transcriptomes of the autotetraploid ‘Xingwang’ and diploid ‘S4’ were compared. We analyzed the key genes of the photosynthesis pathway and the porphyrin and chlorophyll metabolism pathway, and the upregulation of HemB may promote the increased chlorophyll contents of ‘Xingwang’. The upregulation of the related genes whose products encode components of PSII and PSI may enhance the light harvesting of ‘Xingwang’, increasing its light energy conversion efficiency.

Supplemental Information

Supplemental Information 1 The sequences of 9 candidate genes.

Putative gene ID and sequence information.

Click here for additional data file.

Supplemental Information 2 Primers used for the expression pattern of qRT-PCR products.

Putative gene, size, function and primer sequence information.

Click here for additional data file.

Supplemental Information 3 The annotated result of the annotated 44465 unigenes.

Gene ID, and the annotation results of KEGG, GO, NR, NT, Swissprot, Pfam, KOG.

Click here for additional data file.

Supplemental Information 4 24221 DEGs information between the diploid and tetraploid transcriptomes.

Gene ID, S_FPKM, X_FPKM, log2(X/S), Qvalue(S-vs-X) and Pvalue(S-vs-X).

Click here for additional data file.

Supplemental Information 5 The determined of GO classifications.

Gene ID, GO Term, Level 1, Level 2 and log2(X/S).

Click here for additional data file.

Supplemental Information 6 The determined of KEGG classifications.

Gene ID, KEGG Pathway Name, Level 1, Level 2, and log2(X/S).

Click here for additional data file.

Supplemental Information 7 Chromosome observations of stem tip cells of A. buergerianum with different ploidy.

A and a: A. buergerianum X, chromosome number of diploids (2n = 2x = 26); B and b: A. buergerianum S, chromosome number of autotetraploids (2n = 4x = 52). Figures a and b are enlarged views of the red circled cells in Figures A and B.

Click here for additional data file.

Supplemental Information 8 Comparison of the DNA content of the leaves.

A: A. buergerianum S: DNA content of diploids (the main peak at channel 100); B: A. buergerianum X. DNA content of autotetraploids (the main peak at channel 200). The main peaks represent mature cells and the secondary peaks represent meristematic cells.

Click here for additional data file.

Supplemental Information 9 Transpiration rate of X and S in different months.

X: Tetraploid; S: Diploid

Click here for additional data file.

Supplemental Information 10 Diurnal variation of net photosynthetic rates of X and S in different months.

X: Tetraploid; S: Diploid

Click here for additional data file.

Supplemental Information 11 Stomatal conductance changes of X and S in different months.

X: Tetraploid; S: Diploid

Click here for additional data file.

Supplemental Information 12 Diurnal variation of intercellular CO2 concentration in X and S in different months.

X: Tetraploid; S: Diploid

Click here for additional data file.

Supplemental Information 13 Number of gene based on the NR, NT, SwissProt, KEGG, KOG, Pfam, and GO databases.

A total of 44665 Unigenes were annotated, accounting for 86.21% of all Unigenes.

Click here for additional data file.

Supplemental Information 14 Chlorophyll synthesis pathway.

Up-regulated genes in A. buergerianum X are represented by purple boxes, and down-regulated genes are represented by blue boxes.

Click here for additional data file.

Supplemental Information 15 Validation of nine genes by qRT-PCR.

Click here for additional data file.

Supplemental Information 16 E-value Distribution, Species Distribution and Similarity Distribution.

Click here for additional data file.

Supplemental Information 17 Meteorological data of the date of determination of gas exchange parameters.

The experimental date of the gas exchange parameters, the weather situation, temperature and wind and direction.

Click here for additional data file.

Supplemental Information 18 The unigene length distribution.

Click here for additional data file.

Supplemental Information 19 Raw data for gas exchange parameters analysis.

The measurement values of Pn, Gs and Ci of samples X and S in different time periods from May to September.

Click here for additional data file.

Supplemental Information 20 Experimental data of chlorophyll content and chlorophyll fluorescence.

The measured value and average value of chlorophyll a, chlorophyll b, total chlorophyll content; the measured value of chlorophyll fluorescence and the average value of Fv/Fo and Fv/Fm.

Click here for additional data file.

Supplemental Information 21 Statistics on the results of sequencing X and S.

Click here for additional data file.

Supplemental Information 22 qRT-PCR experiment results.

The results were used to verify the expression of candidate genes.

Click here for additional data file.

Supplemental Information 23 Sample RNA test results.

Click here for additional data file.

Supplemental Information 24 Corresponding gene annotation results.

The results of unigenes annotated in the NR, NT, SwissProt, KEGG, KOG, Pfam, and GO functional databases.

Click here for additional data file.

Supplemental Information 25 94,755 results from CL3.Contig1_All to CL4972.Contig2_All.

Aligned to the GO database.

Click here for additional data file.

Supplemental Information 26 94,756 results from CL4974.Contig1_All to Unigene22704_All.

Aligned to the GO database.

Click here for additional data file.

We thank Fenfen Si of Shandong Agricultural University for assistance with the experimental methods.

Additional Information and Declarations

Competing Interests

Author Contributions

DNA Deposition

Data Availability

The authors declare that they have no competing interests.

Yi Wang conceived and designed the experiments, performed the experiments, analyzed the data, prepared figures and/or tables, authored or reviewed drafts of the paper, and approved the final draft.

Bingyu Jia conceived and designed the experiments, performed the experiments, analyzed the data, prepared figures and/or tables, authored or reviewed drafts of the paper, and approved the final draft.

Hongjian Ren performed the experiments, authored or reviewed drafts of the paper, and approved the final draft.

Zhen Feng conceived and designed the experiments, analyzed the data, authored or reviewed drafts of the paper, and approved the final draft.

The following information was supplied regarding the deposition of DNA sequences:

The samples S1, S2, S3, X1, X2, X3 clean reads are available at SRA: SRR12489395 to SRR12489400 and at FigShare:

Wang, Yi; Jia, Bingyu; Ren, Hongjian; Feng, Zhen (2020): Samples cleandata raw data.gz. figshare. Dataset. DOI 10.6084/m9.figshare.12783185.v1.

Jia, Bingyu; Wang, Yi; Ren, Hongjian; Feng, Zhen (2020): Sample X1, X2, X3 cleandata raw data.gz. figshare. Dataset. DOI 10.6084/m9.figshare.12786806.v1.

Use TSA Submission: SUB10222818 to get the assembled UniGenes.

The following information was supplied regarding data availability:

The qRT-PCR experiment results and sample RNA test results are available in the Supplemental Files.

The data is available at FigShare: Wang, Yi; Jia, Bingyu; Ren, Hongjian; Feng, Zhen (2020): Samples cleandata raw data.gz. figshare. Dataset. DOI 10.6084/m9.figshare.12783185.v1.

Jia, Bingyu; Wang, Yi; Ren, Hongjian; Feng, Zhen (2020): Sample X1, X2, X3 cleandata raw data.gz. figshare. Dataset. DOI 10.6084/m9.figshare.12786806.v1.

The raw reads are available at NCBI: SAMN15870859-SAMN15870864 and Bioproject: PRJNA658440.

This Transcriptome Shotgun Assembly project is available at DDBJ/EMBL/GenBank: GJIN00000000. The version described in this article is the first version, GJIN01000000.

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
