# Peer review of "Ploidy level enhances the photosynthetic capacity of a tetraploid variety of Acer buergerianum Miq"

_PeerJ, doi:10.7717/peerj.12620_

## Round 0.1 · original submission · Major Revisions

Your manuscript has now been reviewed by two reviewers and their comments are appended below. I agree with them in finding your manuscript and results interesting, however, they have raised several concerns that need to be addressed by a major revision. Especially, I agree with the concerns about the validity of the findings and the missing details about the methodology and they need to be accurately clarified. All the reviewers' comments are pertinent and should be addressed in detail in the revision. I ask the authors to get editing help from a professional English language correction service or colleagues with full professional proficiency in English.

Reviewer 1 ·

Basic reporting

It would be convenient to review the English. For example Lines 52 to 54: description is redundant and not precise. Line 56: ‘Superior’ is ambiguous.
Please provide citations for every statement in the introduction and discussion. For example Lines 58 to 84 require citations on ‘require more energy and carbohydrates’, ‘changes in the photochemical characteristics of the nucleus’, ‘the photosynthetic rate of plant cells is related to the DNA content of the cells’ and so on after every statement.
64
Regarding the structure: the introduction is not complete as it does not offer full information and citations to understand the background and purpose of the research. Methods are not descriptive enough to repeat the research (details below).
Results are not precise and subjective judgements are included: Examples. Line 188. ‘show that the trends of X and S are consistent’, line 190 ‘The peak transpiration rate of X was greater than that of S’, line 195 ‘an obvious "photosynthetic nap" phenomenon is observed’. Line 197 ‘The daily changes of the net photosynthetic rate (A) of X and S leaves from May to September are the same (Fig. 4).’ and so on.
Also, Differences identified on the chloroplast structure should be described by data. On the discussion, basic information is included that should be on methods section instead, like the one in lines 300, 303, 315.
The connection between the differences identified on the chloroplast and the photosynthetic measurements should be supported by the literature.
Figures: Fig. 3 to 6 do not have statistic information, are differences significant? Fig 1. Please add the 5'->3’ sense of the sequences. There are no figure captions, so figures could not be read for the review. Raw data on photosynthetic study only includes the average values used to build the graphs. There is no raw data then.

Lines 78 to 84: rewrite to clearly state what is the purpose of the study. ‘To reveal the biological characteristics of tetraploid formation’ is a too wide statement; yield is not measured on this study; ‘natural conditions’ are not defined; ‘high light efficiency breeding’ is not defined.
Conclusion is not sustained on data presented: line 415 ‘stronger photosynthesis ability’ needs data to be statistically validated.

Experimental design

The research question here needs to be defined regarding what is the genetic difference between clones ‘S’ and ‘X’ (only chromosome number or also genetic content). Also, the photosynthetic study needs to be justified more in detail.
Description of methods needs to be thoroughly improved.
Raw data and statistical analysis of photosynthetic measurements needs to be provided. A quantitative analysis of differences in chloroplast structure needs to be performed.

- Describe the origin of the plants on methods. How was the polyploid obtained and what is the origin? Is it an autotetraploid or an allotetraploid? This is fundamental as you are comparing a diploid and a tetraploid that, excepting their chromosome number, may have identical or (slightly to very) different genetic content.

- Identify the plants so that the study can be repeatable (Are they part of a botanical collection and have an id name or number?
- Explain the propagation method and how many individuals were used to perform the measurements.
- Photosynthesis parameters have been estimated through punctual measurements. Differences found are punctual and need to be statistically analyzed to verify that they are significant.
- The transpiration rate is not a photosynthetic parameter, please explain why do you include this measurement and what is it relevant for.
- Information provided on methods does not allow to repeat the study (number of plants measured, dimensions of the plants, conditions (field, greenhouse), soil and container, irrigation, temperature humidity and light intensity.
- Define photosynthetic nap and explain why is it relevant for this study.

Validity of the findings

The introduction and discussion should state what is the relevance of the study and findings.
There needs to be a clearer connection between the purpose stated at the end of the introduction, the statements made at the results section, the explanations and analysis provided as a discussion and the study conclusions. Each parameter measured needs to be justified and inter-related with the rest to get to a conclusion.

Additional comments

The topic is relevant for the field and it would be interesting to have this research published. However, the purpose, methods and results of this study need to be worked out to reach a meaningful scientific contribution. I would advise the authors to thoroughly review the manuscript and their research data before further submission of this manucript.

Reviewer 2 ·

Basic reporting

The basic structure of the publication follows the traditions style and has a good flow.
The usage of the English language is adequate.

The manuscript would benefit from having a bot more background to the system. The authors compare a diploid and a tetraploid genotype of Acer buergerianum. What is the natural stage? Was the tetraploid artificially obtained or both occur in nature? Are there any ecological/arquitectural/niche differences?

Figure captions need to be more detailed.

Experimental design

The experimental design needs detail in all sections.
1. Polyploidy identification -
this preparation described does not result in single cells for flow cytometry. How were the cells separated? How many cells were tested?
The dye utilised is not suitable for chromosome counting, but for bacterial and cell wall staining. It is not possible to count the number of chromosomes, and therefore to infer policy level. How many cells were counted? From how many different plants? A lot more detail is required and better images with accurate counts.

2. Diurnal changes in photosynthesis - how many plants were tested? Were they tested during one one day each month? Or do the graphs show an average of all days of the month?

3. Electron microscope sample preparation of leaf chloroplast
The method described is not for electron microscopy. I did a web search and did not see any Pentaflex electron microscope (https://www.microscopes.eu/en/Brand/Pentaflex/). I wonder if the authors meant light microscopy?

4. Transcriptome sequencing
What genotypes were sequenced? How many samples, how many replicates? What was the objective?

Validity of the findings

As the methodology is very vague, so are the results.
1. chromosome number needs to be accurate (or as accurate as possible, using the correct method)
2. flow cytometry - DNA content needs to be expressed as numbers, not as "about twice".
3. photosynthetic analysis - there is a long description, but no analysis was done. There is no conclusion for this experiment
4. microscopy - what is the length of the cells/organeles? How many were measured? Without this information, there is no reasonable way to decide if the findings are typical. Statistical analysis is needed.
5. Transcriptome analyses - this is the part that was least clear. What was being compared? What are the candidate genes and how are they validated? What is the purpose of this experiment? How does it tie with the rest of the study?

Additional comments

The work on tetraploidization is of great interest to the scientific community. Having a system to work on gives you the opportunity to find interesting information that can contribute to the general knowledge. The efforts of the authors are appreciated. However, the work needs to be clearly explained. What was the goal? What is the background of the work?
The manuscript lacks numbers, proper description of method, analyses and description and interpretation fo the findings. I would encourage the authors to revise it carefully.

---

## Round 0.2 · Major Revisions

Your manuscript has now been re-reviewed by one reviewer and the comments are appended below. Your manuscript improved, however, several concerns need to be addressed by a major revision. Especially, I agree with the concerns about the missing details on the methodology and lack of statistical analysis to back the accepted hypothesis that need to be accurately clarified. Moreover, the writing of the manuscript should be improved using a professional English language correction service. All the reviewer comments are pertinent and should be addressed in detail in the revision.

Reviewer 1 ·

Basic reporting

• Clear and unambiguous, professional English used throughout.

The writing should be reviewed carefully as there are confusing sentences and imprecise use of words.

• Literature references, sufficient field background/context provided.

No, the authors should try to shape the introduction so that the research can be understood. Many citations are missing.

• Professional article structure, figures, tables. Raw data shared.

The article structure is not consistent, as there is not correspondence between contents described in introduction (not enough information provided to understand and justify the research), methods (poor descriptions, some parts omitted), results (not properly described, information not well distributed between text and figures/tables), discussion/conclusion (every single content presented on results should be discussed and highlighted on the conclusions.

• Self-contained with relevant results to hypotheses.

It is not clear whether all the contents presented are relevant to reach the conclusions. Authors should try to order contents and justify them.

Experimental design

• Original primary research within Aims and Scope of the journal.

Yes

• Research question well defined, relevant & meaningful. It is stated how research fills an identified knowledge gap.

The question is not well defined as there is misuse of terms, however it is relevant and will shed light into the field.

• Rigorous investigation performed to a high technical & ethical standard.

The statistic analysis and some basic information for repeatability are missing. Authors should be more rigorous on how they present their data and also on how they use that data to reach conclusions

• Methods described with sufficient detail & information to replicate.

No, information is missing

Validity of the findings

• Impact and novelty not assessed. Negative/inconclusive results accepted. Meaningful replication encouraged where rationale & benefit to literature is clearly stated.

Statistic analysis of the gas exchange data needs to be performed to know if the results presented are conclusive about the differences in gass exchange parameters between ploidies in this specie. It would be great if data on growth, chlorophyll content and leaf fluorescence under mild stress could be added to verify that physiology and gene expression are somewhat connected.
Data on morphological measurements should be presented in tables and analyzed statistically. Otherwise the histology should be described in results as qualitative but not quantitative.

• All underlying data have been provided; they are robust, statistically sound, & controlled.

No, as commented before

• Conclusions are well stated, linked to original research question & limited to supporting results.

No. Rewrite the conclusions to be more specific and include all the highlights from different sections describe in the result’s section

• Speculation is welcome, but should be identified as such.

In this sense, authors need to be more careful and specific when they state their results on the discussion section and avoid long paragraphs presenting basic information without a point.

Additional comments

This study intends to investigate the differences between diploid and tetraploid Acer buergerianum in terms of photosynthetic capacity. For that purpose, physiology (different gas exchange parameters), anatomy (chloroplast hystology) and gene expression (differences observed on a massive study validated by PCR methods) are evaluated.

However, results need to be statistically analyzed to reach conclusions, as the graphs and tables show punctual differences in gas exchange parameters and anatomical evidences that could be related to observed differences on gene expression. I would suggest to apply an ANOVA test to results regarding physiology and present hystology results in a quantitative way.

Unfortunately, without the propper statistic analysis, the research manuscript would not be valid for publication.

I hope the comments here and on the pdf file are useful for your team to improve the manuscript.

Annotated reviews are not available for download in order to protect the identity of reviewers who chose to remain anonymous.

---

## Round 0.3 · Major Revisions

Dear authors, unfortunately, it was impossible to contact the two previous reviewers to check your revised manuscript, so we needed to contact a new reviewer. I agree with the reviewer in finding overall your manuscript improved however, the reviewer has raised several concerns that need to be addressed by a major revision. I agree with the comments and changes suggested by the new reviewer and they should be addressed in a revised version of the manuscript. Best regards.

Reviewer 3 ·

Basic reporting

In this manuscript, the authors compared the gas exchange parameters, chloroplast structure, chlorophyll contents, and chlorophyll fluorescence parameters between the tetraploid Acer buergerianum 'Xingwang' and the diploid 'S4', tehn compared the transcriptomes of the autotetraploid 'Xingwang' and the diploid 'S4' of A. buergerianum. They aimed to find out the effect of ploidy level on the photosynthetic capacity and the possible mechanisms underlying the differences. However, major revisions are needed for the current manuscript before acceptance.

Experimental design

Please add some necessary references for some methods.
Line 170-171, Pn, should be Pn, and so did Gs, Ci all through the manuscript. And did you measure transpiration rate? It should always be with the other parameters.

Validity of the findings

Lin 288, should “identification” more suitable for “detection”?
Lin 290-292. It would be better to move the sentence to introduction section or plant materials subsection.
Line 358, line 372. These two subsections should be simplified and most of the details could be presented in supplementary materials instead of figures and tables in main text. For example, Table 5 should be moved to supplementary materials.

Line 399-407, it would be better to do pathway enrichment analysis of different set of genes related with photosynthesis by suing MapMan (Thimm et al., 2004; Lu et al., 2020). And also the the weighted gene coexpression network analysis (WGCNA) to relate photosynthetic indices with trasncriptomic data.

Table 3 and 4 changed to Figure. Fv, should be Fv, and so did the other indices.

Table 6. The table should be in three lines style.
Figure 10 the quality and readable of the figure is low. Can the authors improve the figures. I think there are other sketch of the biosynthesis pathway of chlorophyll.
Figure 11. The quality of the figure is also low. Can the authors draw a figure instead of add the information in a existing picture. There are some other method to present the photosynthesis pathway.
Did hemB presented in results section?

Additional comments

Title:
The title can not cover the content of the main text.

Introduction.
The first and the second paragraph should be reorganized into two part: the first is the effect of ploidy level on photosynthetic ability, and the second should be the possible mechanisms, including chlorophyll content, the structure of chloroplast, the related genes expression, etc. The authors should conclude into several points instead of list the previous studies.
Lin 94: add “with Chinese name of” before “Gong Tong”.

Discussion and conclusions:
It would be better for the authors to present a sketch map for the possible mechanisms.

---

## Round 0.4 · Minor Revisions

Dear authors your manuscript has been revised by one of our reviewers. I agree with the reviewer in finding your article improved and the raised concerns have been addressed. Some minor revisions clearly explained by the reviewer need to be addressed to accept the manuscript for publication.

Reviewer 3 ·

Basic reporting

The manuscript has been revised according to the comments except one point.
1. For abbreviation Pn, P should in italic, n should be in subscript. Please see literature Gao et al., 2017 for the model. So did so did Gs, Ci, Tr and Fv.
Gao S, Yan Q, Chen L, Song Y, Li J, Fu C, Dong M. 2017. Effects of ploidy level and haplotype on variation of photosynthetic traits: novel evidence from two Fragaria species. PLoS One 12: e0179899. DOI: 10.1371/journal.pone.0179899

2. The number of tables and figures are too many for a paper. Please merge small pictures into large ones, or move some to the supplementary materials or simplify them. For example, Figure 10 and Figure 11 can be merged into one figure with a and b. So did Figure 7 and Figure 8. Table 1 and Figure 12 can be moved to supplementary materials.

Experimental design

good.

Validity of the findings

good.

---

## Round 0.5 · Minor Revisions

The authors answered the comments from the reviewer. Before accepting the paper for publication, the authors need to deposit the assembled UniGenes in a public repository. Moreover, the corresponding annotations need to be included in the paper or in the repository. This is important to make the data public available to the scientific community.

---

## Round 0.6 · Minor Revisions

Dear authors, after revision of an additional reviewer, your manuscript still requires changes.

In the revised version of the manuscript there is mention that data has been uploaded somewhere, but there is no mention of where and how to obtain it. There are 51815 GeneIDs listed for Unigenes and Cluster, but no sequence presented to represent them. Likewise there is no attachment of the annotations highlighted such as GO or KEGG except for the description. Links to the data and understanding of the data need to be presented to relay and understanding of the data interpreted to the reader.

Appropriate margin headings appear to be missing from all supplemental figures making it difficult to interpret context and content. “Fluorescence” is the correct spelling in S2. Figure S7 highlights the numbers of unigenes; however, no unigene data is provided as supplement nor third-party resource.

The manuscript is still in need of revisions.

EDITS
LINE NO: / BEFORE / AFTER / [COMMENTS]
LINE 45: / 200-300 / ? / [Is this correct if average value is 5262?]
LINE 48: / 24221 / . / [These sequences at least need to be available and with accompanying annotations in a separate table.]
LINE 76: / ploides / ploidy / [.]
LINE 255: / All-unigene sequences were assigned GO terms / . / [Neither sequence nor annotations available!]
LINE 273: / Nine candidate genes / . / [No sequences available, except primers.]
LINE 299: / Dole el et al. / ? et al. / [citation not present.]
LINE 365: / 42,804,595 nt and 44,971,637 nt raw data / . / [not available!]
LINE 367: / 51807 unigenes / . / [not available!]
LINE 372: / 44465 unigenes were annotate / . / [noot available!]
LINE 375: / (Fig. S7). / . / [Where is the accompanying table?]
LINE 379: / total of 24221 DEGs were screened / . / [not available.]
LINE 382: / GO and KEGG classifications were determined / . / [not available, nor sequences.]
LINE 457: / Researchers have shown / Research has shown / [.]
LINE 471: / we present a sketch map / we present a schematic diagram / [.]
LINE Figure 5: / sketch map / schematic diagram / [.]"

---

## Round 0.7 · accepted · Accept

The authors answered the comments.

Please ensure the assembled unigene is made public as soon as possible.